# Predicting Aneurysmal Degeneration in Uncomplicated Residual Type B Aortic Dissection

**DOI:** 10.3390/bioengineering11070690

**Published:** 2024-07-08

**Authors:** Arianna Forneris, Ali F. Hassanabad, Jehangir J. Appoo, Elena S. Di Martino

**Affiliations:** 1Department of Biomedical Engineering, Schulich School of Engineering, University of Calgary, Calgary, AB T2N 1N4, Canada; arianna.forneris@ucalgary.ca; 2R&D Department, ViTAA Medical Solutions, Montreal, QC H2K 1M6, Canada; 3Section of Cardiac Surgery, Department of Cardiac Sciences, Libin Cardiovascular Institute, University of Calgary, Calgary, AB T2N 1N4, Canada; ali.fatehihassanabad@ahs.ca (A.F.H.); jappoo@ucalgary.ca (J.J.A.)

**Keywords:** aortic dissection, aneurysmal degeneration, hemodynamics, CFD

## Abstract

The formation of an aneurysm in the false lumen (FL) is a long-term complication in a significant percentage of type B aortic dissection (AD) patients. The ability to predict which patients are likely to progress to aneurysm formation is key to justifying the risks of interventional therapy. The investigation of patient-specific hemodynamics has the potential to enable a patient-tailored approach to improve prognosis by guiding disease management for type B dissection. CFD-derived hemodynamic descriptors and geometric features were used to retrospectively assess individual aortas for a population of residual type B AD patients and analyze correlations with known outcomes (i.e., rapid aortic growth, death). The results highlight great variability in flow patterns and hemodynamic descriptors. A rapid aortic expansion was found to be associated with a larger FL. Time-averaged wall shear stress at the tear region emerged as a possible indicator of the dynamics of flow exchange between lumens and its effect on the evolution of individual aortas. High FL flow rate and tortuosity were associated with adverse outcomes suggesting a role as indicators of risk. AD induces complex changes in vessel geometry and hemodynamics. The reported findings emphasize the need for a patient-tailored approach when evaluating uncomplicated type B AD patients and show the potential of CFD-derived hemodynamics to complement anatomical assessment and help disease management.

## 1. Introduction

In aortic dissections (AD) classified as type B, intimal tearing occurs distally to the left subclavian artery (LSA) and extends into the descending aorta for a prognostic profile characterized by a range of outcomes and management options based on initial presentation [1]. Uncomplicated type B ADs are commonly treated with medical (pharmacological) therapy aimed at stabilizing the patient and delaying disease progression but with no success in inducing favorable remodeling and preventing adverse long-term outcomes. Different studies have tried to address the lack of clinical consensus on whether an early intervention could be beneficial for some patients, and generally on how and when to opt for intervention [2,3,4]. To this day, the optimal treatment choice for stable type B AD is still debated [5].

The formation of an aneurysm in the false lumen (FL) is a common long-term complication for type B AD and significantly contributes to late mortality. The rapid aortic growth behind aneurysmal degeneration in the FL is not fully understood and is likely to be affected by local hemodynamics along with the presence of pre-existing conditions and comorbidities. According to current guidelines, a persistent FL perfusion favoring patency is regarded as an independent indicator of risk for late adverse outcomes and aneurysmal degeneration, especially when combined with a maximum aortic diameter greater than 40 mm [6,7,8,9]. The morphology and hemodynamics of individual aortas, however, greatly affect different aspects of the pathology, such as initiation, progression and development of later complications as well as the success of treatment procedures. Given the complex nature of non-physiological changes introduced by an AD from a geometric and hemodynamic perspective, high inter-patient variability is expected to be one of the causes of undefined treatment guidelines for this subgroup of patients [10,11]. For this reason, the investigation of patient-specific hemodynamics seems promising in the context of finding a patient-tailored approach to improve prognosis and disease management for uncomplicated type B AD patients.

Wall-shear stress (WSS) is an important parameter in the analysis of aortic wall diseases, as it plays a major role in promoting structural and compositional changes that affect the wall’s mechanical properties, especially in the presence of altered fluid dynamics. Regarding AD, high WSS levels have been linked to the initial delamination and tear formation between the wall layers [12] and are thought to contribute to the retrograde expansion of the tearing and the formation of retrograde type A dissection, known to be a complication in type B patients [13]. Imaging modalities, such as CT, 3D, or 4D-MRI, can provide anatomical and flow information, but provide limited resolution for the near-wall regions, which is essential for the estimation of WSS, and none offer easy access to non-invasive pressure measurements for the dissected area and FL.

In recent years, several investigations made use of computational fluid dynamic (CFD) simulations to clarify the role of fluid dynamic and anatomic parameters in the late aneurysmal degeneration of uncomplicated type B ADs. Some of these studies, such as Karmonik et al., 2012a [14] and Tse et al., 2011 [15], aimed to prove the applicability of CFD methods to ADs and their potential to predict flow patterns. Tse et al. reported higher time-averaged wall-shear stress (TAWSS) in the true lumen (TL) compared to the FL and suggested a pressure difference between the FL and TL in the pre-aneurysmal geometry to be indicative of the site of dilatation. Similarly, Chen et al., 2013a [16] described the flow exchange between the false and true lumens in a patient-specific type B AD, reporting high WSS around the entry tear. The authors also compared the results of laminar and turbulent models in the same geometry concluding that laminar and turbulent simulations presented similar flow patterns. Cheng et al., 2010 [17] studied a single case of acute type B AD for a detailed investigation of patient-specific hemodynamics. Their findings included elevated oscillatory shear index (OSI) in the FL, elevated TAWSS at the entry tear, higher pressure in the TL compared to the FL in the proximal descending aorta, and an opposite situation in the distal district. In the same publication, the effects of the Newtonian assumption for blood modeling are explored and quantified as minor differences in WSS and pressure estimation. This group’s research also focused on investigating the effect of AD morphological features on hemodynamics: Cheng et al., 2013 [18] compared four type B AD patients (two complicated and two uncomplicated) in terms of tear height, width, distance from aortic arch and percentage of flow in the FL. Despite limitations, this study demonstrated a significant relationship between FL flow and entry tear size and location, with larger flows associated with larger and more proximal tears.

Despite the high mortality rate, AD disease generally has a low incidence, making it challenging to achieve large population studies. Only a few publications were able to report on multi-patient studies. Cheng et al., 2015 [19] applied their methods to a population of four medically treated (uncomplicated) and four TEVAR (complicated) type B AD patients. The authors were able to confirm their previous findings on tear morphology and investigate the pressure in the FL as a possible contributor to aneurysm formation. Another multi-patient study from Shang et al., 2015 [20] focused on a population of fourteen acute type B AD patients with either stable diameter or rapid aortic growth and correlated CFD results with known outcomes. Their findings associated increased TAWSS and larger FL flow proportion with rapid FL expansion. More recent studies seem to confirm these results. Zhu et al., 2021 [21] showed significant differences in terms of the number of re-entry tears and true/false lumen pressure difference for patients with stable aortic size versus patients with aortic dilation, while Shad et al., 2022 [22] reported a strong correlation between false lumen growth and fenestration flow velocity.

The amount of literature on CFD applied to the prediction of aneurysmal degeneration in type B AD is limited and mostly offers preliminary insights and findings, but it also emphasizes the potential of this method to assist clinical decisions and disease management.

The present study proposes the use of CFD-derived hemodynamic descriptors to retrospectively assess individual aortas for a population of type B ADs, and analyze potential correlations with known outcomes, such as rapid aortic growth indicating aneurysmal degeneration.

## 2. Materials and Methods

### 2.1. Study Design

Ethics approval was obtained by the University of Calgary—Conjoint Health Research Ethics Board (CHREB) for a retrospective study on a population of consecutive patients presenting residual type B aortic dissection after receiving hemi-arch repair for a type A aortic dissection between 2009 and 2014. Given the retrospective nature of this study, only patients with at least one-year follow-up, or urgent reintervention within a year, were selected, for a total of forty-one subjects. The following exclusion criteria were then applied: CT scans with poor contrast/resolution, extensive presence of thrombus in the descending aorta, marked intimal flap motion when detrimental to image quality and identification of anatomical features, presence of intramural hematoma or penetrating aortic ulcer, connective tissue disorders (i.e., Marfan syndrome and Loeys–Dietz syndrome), other underlying conditions (i.e., giant cell aortitis), lack of immediate scan after surgery for type A AD, and previous intervention in the descending aorta. The final study population included twenty-two subjects with residual type B aortic dissection.

To assess aortic diameter and monitor aortic growth on follow-up scans, three planes were defined on axial images along the artery: proximal descending aorta above the left pulmonary artery, mid-descending aorta at the level of the left inferior pulmonary vein, and distal descending aorta at the diaphragm level. The population was divided into two groups based on rapid aortic growth (3 mm/year) characterizing at least one of the three locations, or stable aortic size, according to clinical guidelines [23]. Among the stable-size patients, two subgroups were then identified based on a favorable or adverse (death) long-term outcome.

Patient-specific aortic anatomies were extracted from CT images (spatial resolution 0.703 × 0.703 × 0.625 mm) obtained after surgery in the ascending aorta, representing the initial observation time, and 3D geometries were reconstructed using the commercial image-processing software Simpleware ScanIP 2016.09 (Synopsys, Inc., Mountain View, CA, USA). Final 3D geometries were inclusive of the ascending aorta, aortic arch with supra-aortic branches (brachiocephalic artery (BCA), left common carotid artery (LCCA), and left subclavian artery (LSA)), and descending aorta (DA), while smaller aortic ramifications were disregarded (Figure 1). After reconstruction, the triangulated surface of the aortic lumen was imported to ICEM 17.0 (ANSYS, Canonsburg, PA, USA) for discretization of the computational domain into tetrahedral elements. The mesh also consisted of prismatic elements refining the near-wall region for increased accuracy. A sensitivity analysis was performed to ensure the selection of a mesh size able to produce accurate results at a reasonable computational cost. Consecutive grid refinements were used to obtain a range of mesh sizes from coarse (approximately 1 million elements) to fine (approximately 5 million elements), for a total of ten different meshes for each case. Steady-state simulations were then run at the cycle-averaged flow rate to obtain mean and maximum WSS values for each mesh. The effect of mesh density on the variables of interest was assessed by means of relative percentage difference with respect to the finer (more accurate) mesh. The results from a representative sensitivity analysis are reported in Figure 2: the effect of mesh refinement on the mean and maximum WSS was negligible for a mesh density of about 3 million tetrahedral elements. The final mesh density for the population ranged between 3 and 5 million elements depending on individual aortic size and morphology.

### 2.2. Computational Simulations

#### 2.2.1. Boundary Conditions

Due to the lack of patient-specific information for our study population, realistic boundary conditions were obtained from phase-contrast magnetic resonance imaging (PC-MRI) data of a thoracic aorta and adapted for each AD simulation. Flow rates were extracted at the MRI acquisition plane across the thoracic ascending and descending aorta and used to evaluate the flow loss through the supra-aortic branches. According to the information derived from imaging, about 55% of the total time-averaged aortic flow exited the domain through the descending aorta, while the remaining 45% was diverted into the three main arch branches. Although the percentage of flow entering each one of the branches is measured to be 5–7% in a healthy aorta [24], the downstream resistance in an AD geometry is likely to increase due to the reduction of the TL cross-section with a consequent higher proportion of flow to the branches [25].

A flat time-varying velocity profile was applied at the inlet of each geometry based on the flow rate waveform in the ascending aorta and consistently with the cardiac cycle period of 0.8 s (76 bpm heart rate) (Figure 1). Outflow boundary conditions were defined at the domain outlets by imposing the flow distribution previously evaluated. The flow division among the three arch branches was defined based on the cross-sectional area of each arch outlet and, consequently, adapted for each patient-specific geometry. The arterial wall was assumed to be rigid, and a no-slip boundary condition was applied.

#### 2.2.2. Blood Rheology and Fluid Flow Regime

The blood was modeled as an incompressible Newtonian fluid with a density of 1060 Kg/m^3^ and a dynamic viscosity of 0.00319 Pa·s. Simulations run for this study employed a laminar flow model, a commonly accepted simplification for large vessels where the average velocity results in relatively low Reynolds numbers. The inlet peak Reynolds number and critical Reynolds number were measured for each geometry, based on the Womersley and Strouhal numbers [26]. The peak Reynolds number at the inlet was found to range between 3030 and 4655, while the critical Reynolds number varied between 4366 and 5552, resulting in subcritical inlet flows for all the analyzed geometries. Based on these numbers, along with the reported minor effects of turbulence on simulation results [25], the assumption of laminar flow is considered reasonable for this study.

#### 2.2.3. Numerical Implementation

Transient time simulations were run in FLUENT 17.0 (ANSYS, Canonsburg, PA, USA) to simulate the blood flow during the entire cardiac cycle. A SIMPLE scheme was employed for the pressure–velocity coupling algorithm and the Navier–Stokes equations were temporally and spatially discretized by means of a second-order implicit transient formulation and a second-order upwind scheme, respectively. Preliminary steady-state simulations were performed at the cycle-averaged flow rate and employed for mesh sensitivity analysis and time step selection. A fixed uniform time-step size (in the range of 0.0005–0.002 s) was selected on a patient-specific basis according to the mesh size and maximum velocity in individual geometries (Courant–Friedrichs–Lewy condition). All simulations consisted of four cardiac cycles to ensure complete development of the velocity field and attain a periodic solution before exporting the variables of interest. WSS-based hemodynamic wall descriptors (HWDs), namely TAWSS, OSI, and relative residence time (RRT), were computed from simulation results of each geometry and evaluated for the descending aorta, as a region of interest for late aneurysmal degeneration, and separately for the false and true lumens and the area surrounding the entry tear margins.

### 2.3. Statistical Analysis

Statistical analysis was performed in MATLAB 8.3.0.532 (R2014a) (The MathWorks Inc, Natick, MA, USA). A Fisher exact test was used to compare categorical variables. Continuous variables were tested for normality by means of Shapiro–Wilk test, and apposite transformations were applied in case of non-normal data distribution. Unpaired *t*-tests or Mann–Whitney tests were then run appropriately to compare the groups. Potential relationships between variables were investigated by means of Pearson’s correlation coefficient. Significance was tested for two-tailed *p*-values < 0.05.

## 3. Results

### 3.1. Study Population

Growth analysis was performed on aortic diameter at the three defined locations on follow-up images: Eleven patients (50%) were found to be rapidly growing (n = 11, maximum growth rate 4.61 ± 1.82 mm/y—radiologic surveillance 3.50 ± 1.63 years) and eleven presented a stable aortic size during follow-up (n = 11, maximum growth rate 1.34 ± 0.92 mm/y—radiologic surveillance period 3.41 ± 1.67 years). In the stable-size group, seven patients had favorable long-term outcomes (n = 7—radiologic surveillance 3.47 ± 1.12 years) while four patients, despite showing stable aortic diameter, presented adverse long-term outcomes (n = 4, two re-interventions and four deaths—radiologic surveillance 3.30 ± 2.60 years). Thus, two subgroups were identified as favorable and with adverse long-term outcomes. In the rapid-growth group, seven patients needed re-intervention involving distal arch and/or descending thoracic aortic (or thoraco-abdominal) replacement; four of these patients died during follow-up. Overall, re-intervention was needed for nine patients in the study population (41%) while death was an outcome in eight of the twenty-two cases (36%).

### 3.2. Anatomical Features of the Dissection Aorta

At initial presentation, defined by the most immediate follow-up scan after repair of the ascending aorta, the rapidly expanding and stable aortas presented similar maximum diameter (stable size 35.19 ± 11.10 mm—rapid growth 35.46 ± 7.54 mm; *p* = 0.237). The initial true lumen size was evaluated as ratio of total descending aortic area (TL/DA averaged over the three previously described locations) and was found to be significantly smaller in patients subject to aortic enlargement (stable size 0.42 ± 0.12—rapid growth 0.29 ± 0.11; *p* = 0.009), intrinsically providing an evaluation of the false lumen size as significantly different between the two groups.

Entry tear height (distance between the upper and lower edges) and width (maximal horizontal tear size) were used to estimate the tear area, approximated as a circle. The two groups did not present significant differences in terms of entry tear area (stable size 144.78 ± 173.67 mm^2^—rapid growth 210.01 ± 182.63 mm^2^; *p* = 0.301). The stable-size group presented more re-entry tears on average but did not show a significant difference when compared to the rapid-growth group (stable size 2.1 ± 1.6—rapid growth 1.5 ± 1.4; *p* = 0.415).

Finally, the tortuosity of the aorta was evaluated as a ratio between the length of the vessel (measured along its centerline) and the distance between the origin and end points, respectively, at the inlet and outlet of the geometry. High tortuosity was reported in most of the subjects with no distinct difference between the stable-size and rapid-growth groups (stable size 1.38 ± 0.29—rapid growth 1.38 ± 0.34; *p* = 0.995).

The two subgroups defined according to the long-term outcome of subjects in the stable-size group did not differ significantly in initial maximum diameter (favorable outcome 32.49 ± 4.63 mm—adverse outcome 39.92 ± 17.92 mm; *p* = 0.527), true lumen size as a ratio of total aortic size (favorable outcome 0.40 ± 0.11—adverse outcome 0.46 ± 0.13; *p* = 0.472), or number of re-entry tears (favorable outcome 2.4 ± 1.5—adverse outcome 1.5 ± 1.5; *p* = 0.461) but the entry tear area was different in the two subgroups, with patients that presented adverse outcomes showing a significantly larger entry tear (favorable outcome 67.12 ± 49.27 mm^2^—adverse outcome 280.69 ± 238.70 mm^2^; *p* = 0.015). Aortic tortuosity was non-significantly different between the two stable-size subgroups (favorable outcome 1.30 ± 0.18—adverse outcome 1.52 ± 0.42; *p* = 0.274); however, when the overall study population was assessed according to death or survival, aortic tortuosity was found to be significantly higher in patients who died during follow-up (1.58 ± 0.34 versus 1.26 ± 0.22; *p* = 0.017). Table 1 and Table 2 provide a summary of the main geometric parameters for the two aortic growth groups and the two stable-size subgroups (favorable versus adverse long-term outcome).

### 3.3. False Lumen Flow

The FL flow was evaluated at the systolic peak and expressed as percentage of the total flow entering the descending thoracic aorta. Differences between the rapid-growth and stable-size groups were non-significant (stable size 55% ± 20%—rapid growth 63% ± 15%; *p* = 0.331). Similarly, the two stable-size subgroups did not show significant difference in FL flow (favorable outcome 50% ± 16%—adverse outcome 66% ± 24%; *p* = 0.265), although a trend could be observed towards higher FL flow for the adverse-outcome subgroup. Statistical significance was reached when the entire population was divided according to death or survival: patients who died during surveillance presented a significantly higher flow estimated in the FL (61% ± 18% versus 33% ± 4%; *p* < 0.001).

Possible relationships between geometric features and flow in the FL were explored; a moderate correlation was found between the entry tear area and the percentage of flow in the FL (r = 0.55, *p* = 0.007) and between the true lumen to aortic area ratio and the flow in the FL (r = −0.64, *p* = 0.001).

### 3.4. Flow Patterns and Hemodynamic Wall Descriptors

The complex anatomical changes, introduced by the presence of a dissection flap, generate a considerable flow pattern variability in the descending region of the aorta. High-velocity flows were predominant in the ascending aorta and at narrowed areas in the TL. Flow through the entry tear was often found to be accelerated causing jet impingement on the opposite FL wall. Therefore, disturbed and recirculating flow patterns characterized by low velocities were observed near the posterior FL wall and retrograde dissection.

Flow patterns can be appreciated from streamlines at systolic peak, systolic deceleration, and diastolic peak for four representative cases in Figure 3: rapid aortic growth was reported for patients 1 and 3, with marked difference in their initial maximum diameter (56.88 mm and 29.50 mm, respectively), while patients 12 and 19 showed stable aortic size and presented favorable and adverse clinical outcome (re-intervention and death during two-year follow-up), respectively. A moderately organized and fast TL flow was found in all cases at the systolic peak in opposition to the lower velocities in the FL. The effect of geometry on arterial flow is particularly evident in patient 1: the bending and narrowing in the distal ascending and descending aorta generated local flow disturbances and recirculation during diastole. Patient 3 is a good example of accelerated flow through the entry tear and slow recirculating flow in the retrograde FL during all reported phases, with the diastolic peak being particularly relevant for the formation of complicated and disturbed flow structures in both lumens. Patient 12 had a similar tear configuration to patient 1 (distal to the origin of the LSA); however, the FL flow showed slightly higher velocities with the presence of vortical structures. For patient 19, the flow velocity in the FL and TL appeared to be more similar: the blood is accelerated through a narrower region in the distal FL and forced back into the TL through a re-entry tear, causing higher velocities in the distal TL.

A detailed HWD distribution for the four selected cases is reported in Figure 4, while Figure 5 shows the TAWSS luminal distribution for all the patients in the study population. Elevated TAWSS was found at the entry tear location and narrowed areas of the TL. TAWSS in the FL was found to be uniformly low, except for a region subject to jet impingement caused by flow acceleration through the entry tear (patient 3) or narrowing in the distal descending aorta (patients 1 and 19). High OSI characterized different locations where the flow was subject to frequent directional changes over the cardiac cycle. As the FL is particularly exposed to flow reversal or recirculation, the OSI was generally higher in this region. Areas surrounding the entry tear, mainly exposed to high TAWSS, presented low OSI, except for geometry in patient 1. In this case, the blood accesses the FL through a large tear past the LSA origin and is subject to low velocities, flow recirculation, and re-attachment due to the bulging in the proximal region of this lumen (Figure 3). Patient 1 also presented high OSI at the anterior TL surface where a recirculation area is present at diastole, likely because of the coarctation at the arch origin and the predominant disturbance induced in this area by the highly recirculating flow at the FL entrance. Elevated RRT values characterized the retrograde expansion of the FL in patients exhibiting one (e.g., patients 3 and 12) or in areas of the FL with particularly slow and disturbed flow, such as the bulging area near the entry tear in patient 1.

Trends and statistically significant differences were observed from the separate assessment of the two lumens for all patients: the TL presented a trend towards higher mean TAWSS (1.09 ± 0.68 Pa vs. 0.76 ± 0.42 Pa; *p* = 0.062), lower mean OSI (0.25 ± 0.09 vs. 0.29 ± 0.05; *p* = 0.051), and a significantly lower mean RRT (8.10 ± 10.66 Pa-1 vs. 136.11 ± 582.32 Pa-1; *p* = 0.004).

The dissection groups did not differ significantly with respect to the HWDs; however, when considering the area surrounding the entry tear, a trend was noted towards higher peak TAWSS for the subgroup with adverse outcome among the stable-size patients (favorable outcome 6.73 ± 4.52 Pa—adverse outcome 10.10 ± 4.13 Pa; *p* = 0.251). From this observation, the peak TAWSS was evaluated at the entry tear by considering it distributed over the entry tear margins (entry tear perimeter): this parameter was significantly higher for the adverse-outcome subgroup (favorable outcome 143.44 ± 91.31 Pa∙mm—adverse outcome 452.01 ± 254.30 Pa∙mm; *p* = 0.022). Similarly, when the assessment of this parameter was extended to the entire population, patients reported dead during follow-up presented a significantly higher TAWSS over the entry tear margins (419.08 ± 237.62 Pa∙mm vs. 229.94 ± 153.79 Pa∙mm; *p* = 0.036). The two groups defined according to aortic growth did not show significant differences in terms of HWDs; of note, a trend was observed towards higher TAWSS over the tear margins for the rapid-growth group (stable size 255.65 ± 220.54 Pa∙mm—rapid growth 341.78 ± 189.12 Pa∙mm; *p* = 0.134). Table 1 and Table 2 provide a summary of the main hemodynamic parameters for the groups and subgroups.

### 3.5. Pressure in the Two Lumens

The pressure distribution was evaluated for the two lumens, and differences were obtained as average FL pressure minus average TL pressure on cross-sectional planes perpendicular to the vessel’s centerline along the descending aorta of the four representative cases. Figure 6 shows pressure differences at three time points corresponding to systolic peak, mid-systolic deceleration, and diastolic peak. The difference is expressed as FL pressure minus TL pressure; therefore, a positive pressure difference is indicative of a higher FL pressure compared to the TL.

Slightly higher FL pressure was commonly found during diastole, with the other phases presenting more variability. Patient 1, from the rapid-growth group, was the only case with consistently higher pressure in the FL during both systolic deceleration and diastole. A peak of higher FL pressure was found at the three time points at the level of the entry tear, distally to the LSA. Patient 3 (rapid growth) showed a more complex pressure distribution, particularly during the systolic phase (both peak and deceleration), with peaks of higher FL pressure at the proximal descending aorta (entry tear location) and distally along the vessel. Patient 12, from the stable-size group (favorable outcome subgroup), showed a consistently higher TL pressure during systolic peak and deceleration, with the FL exhibiting marginally higher pressure at the entry tear location. Patient 19, from the stable-size/adverse-outcome subgroup, was found to have a large pressure difference between lumens, with higher FL pressure persisting over the distal descending aorta.

## 4. Discussion

The present study focused on a population of residual type B ADs with highly heterogenous anatomies, likely affected by the hemi-arch repair surgery following type A dissection. The complex geometry of dissected aortas is a crucial aspect that may influence the disease progression as well as the short- and long-term individual outcomes. Specifically, large maximum aortic diameter (>40 mm) [7] and a patent false lumen have been regarded as predictors of late outcome and aortic growth for chronic uncomplicated type B AD [8,9]. In the present analysis, each geometry was evaluated according to initial diameter, true lumen to total aortic area ratio, primary entry tear area, and vessel tortuosity. According to clinical guidelines, three patients (patients 1, 12, and 22) would be considered at risk for aneurysmal degeneration based on the maximum diameter criteria; however, rapid aortic growth was reported for only one of these cases. The initial diameter was overall unable to differentiate the groups under investigation.

The TL/DA area ratio was significantly lower in patients with rapid aortic expansion, suggesting a larger FL, likely to maintain patency and compress the TL, as the distinguishable feature of this group. FL patency is, on the other hand, influenced by the amount of flow entering this newly formed lumen: higher flows are expected to maintain a clear lumen while the presence of low velocities and recirculation may contribute to platelet activation and thrombus formation. By directly influencing the primary communication between the two lumens, the dimension of the entry tear is expected to affect the flow entering the FL. The effects of anatomical factors on flow division were demonstrated by statistically significant correlations found between tear size and FL flow (positive correlation) and between TL/DA area ratio and FL flow (negative correlation). The stable-size and rapid-growth groups, as well as the two subgroups defined from the stable-size patients, did not differ significantly in terms of FL flow or entry tear size. However, from the analysis of FL flow with respect to patients’ survival for the overall population, a significantly higher FL flow was associated with death.

Finally, high tortuosity was commonly found to characterize the aortic geometries with no discriminatory impact between groups defined on growth or between subgroups, despite a trend towards higher tortuosity for the adverse-outcome subgroup. When the entire population was evaluated with respect to death or survival, patients whose follow-up reported death showed significantly higher tortuosity. This finding points to aortic tortuosity, likely the result of axial growth, as an indicator of adverse outcome.

The hemodynamics of AD have been reported to be very complex, with high levels of disturbance promoted by flow recirculation and separation [14,15,18,27]. For this study, flow features in the dissected descending aorta presented high variability among individual anatomies; however, a few common patterns were observed. A generally more disturbed flow characterized the FL, with low velocities and recirculation present at the retrograde portion and jet impingement at the wall facing the entry tear. In contrast, the TL exhibited a more organized, high-velocity flow. Narrowed and bent regions, due to high tortuosity in pathological aortas, also influence the vessel’s hemodynamics and generate areas of accelerated flow (e.g., patient 1). WSS-based descriptors were analyzed separately for the true and false lumens. The need for a separate evaluation of the two lumens stems from the understanding of the effects of local WSS on endothelial function and remodeling. Despite the reciprocal influence between TL and FL, rapid growth and aneurysmal degeneration are likely to be a consequence of disturbed FL flow and WSS acting on the FL wall. While elevated TAWSS values have the potential to damage the endothelium, with consequent tear initiation or enlargement [14,28,29], disturbed flow characterized by low, oscillatory WSS has been reported to have a role in aneurysm formation and rupture [30,31,32]. Given the changes in aortic wall composition that follow delamination and dissection, it is unclear how the newly formed lumen responds to shear stresses. The natural history of aortic dissection, however, includes endothelialization of the FL at chronic stages, as reported from ex vivo studies [33], suggesting a similar pathological process to aneurysm formation may be involved in the aneurysmal degeneration of the FL. In this study cohort, low oscillatory WSS was found in the FL demonstrating a generally more disturbed flow compared to the TL, while high TAWSS levels were observed around the entry tear, as a consequence of flow acceleration in the passage from the TL to the FL. Elevated RRT values were reported at the retrograde extension of the FL as a result of blood recirculation and stagnation [19], likely to favor thrombosis following platelet deposition in this area as observed on follow-up images in a few cases. The role played by thrombus formation in the long-term outcome of the disease is still debated, as partial FL thrombosis has been linked to poor clinical sequels [34].

The computed HWDs did not provide a means to differentiate the aortic growth groups or the outcome subgroups; however, a trend towards higher peak TAWSS at the entry tear region, statistically significant for the adverse-outcome subgroup, prompts some thought-provoking observations. The tear size and shape influence the TAWSS at the tear, with a smaller tear generating higher TAWSS that may promote its enlargement. This aspect also depends on the history of individual ADs: as the entry tear enlarges the TAWSS around it will likely be reduced. Therefore, a small entry tear with high TAWSS at the initial time may evolve into a larger tear, while a large tear with low TAWSS could represent a different point in time of an initially small entry tear with high TAWSS. This is generally true under the assumption of a circular-shaped entry tear, while different shapes and configurations may induce high TAWSS even in large tears. These considerations gain importance when analyzing a group of subjects for which the initial time was surgery for type A and not the formation of the type B dissection. For our population, the peak TAWSS distributed over the tear margins tended to be higher for patients with rapid growth and significantly higher for patients with reported death, in both the stable-size subgroup (adverse-outcome subgroup) and the larger adverse-outcome group defined for the entire population.

The pressure difference between the two lumens may contribute to the compression and dynamic obstruction of the TL, potentially causing reduced flow to the downstream organs, with increased risk for ischemia [35] and likely associated with FL expansion [19]. The compression of the TL, by means of systemic pressure in the FL, is commonly observed at diastole and it could promote FL expansion when large differences are present and maintained during the cardiac cycle. For the study population, markedly higher FL pressure was found at locations of entry tear for the patients whose FL progressed to dilatation during follow-up (e.g., patients 1 and 3). Patient 1 showed higher FL pressure along the entire descending aorta for most of the cardiac cycle (systolic deceleration and diastole); interestingly, this patient presented a large initial diameter (56.88 mm) and late adverse outcome including re-intervention and death. Patient 3 had a more complex distribution in terms of pressure difference, with peaks showing higher FL pressure at the mid-descending aorta and distal descending aorta; of note, the higher growth rate for this patient was found at the mid-descending aorta (at the level of the inferior left pulmonary vein on axial images) and marked intimal flap motion was observed at the mid-descending aorta and distal descending aorta. A similar observation was derived for patient 19 (stable-size group/adverse-outcome subgroup) who showed markedly higher FL pressure at the distal DA close to the re-entry tear and in the same region where movements of the intimal flap were detected on images. While a higher TL pressure is likely to protect the lumen from compression, a higher FL pressure may be a factor in the complicated and multifactorial scenario of aneurysmal degeneration and adverse late outcome in type B AD.

### 4.1. The Adverse-Outcome Subgroup

Despite the small sample size, some results could be derived from the analysis of the subgroups and the patients that had adverse long-term outcome with no reported rapid growth. The adverse-outcome subgroup presented significantly larger entry tear size and peak TAWSS distributed over the entry tear margins when compared to patients from the favorable outcome subgroup. This finding points to the entry tear as one of the key geometric features influencing the clinical progression of AD.

As mentioned, the results obtained for these patients are likely affected by the small sample size; however, the evaluation of individual cases allowed for interesting observations that require separate discussion. Patient 22 had a highly tortuous geometry with a maximum diameter of 66.77 mm, 98% of aortic flow entering the FL at the initial time, and a follow-up of only five months (0.43 years) before re-intervention and death. This patient potentially represents a case of dissection at a more advanced stage compared to other subjects in the study. This hypothesis is confirmed by initial CT images showing a very defined AD geometry with clear thrombus formation partially occluding the already dilated FL, calcifications at the wall, and no flap motion.

CT images for patient 19 and patient 21, obtained shortly after surgery for type A dissection, demonstrated hypointense regions in the FL manifesting no contrast enhancement. Low-intensity areas, commonly found in the FL on AD images, are likely to be associated with very slow and recirculating blood flow with the potential to induce platelet deposition and thrombus formation. For the two specific cases, however, the especially dark, low-intensity regions could indicate an already active thrombus formation in the FL that may have played a role in leading to the adverse outcome during follow-up. In the literature, a partially closed FL has been associated with increased mortality when compared to a patent FL [34], while it is still unclear whether partial thrombosis constitutes a risk factor for aortic enlargement [36,37]. Finally, patient 20 was also a distinct case: the reported increase in tortuosity at follow-up is likely associated with axial aortic growth although this could not be measured based on diameter assessments on axial images.

The need to define two subgroups not only highlights the complex variability among residual type B dissection patients and outcomes but poses important questions on whether a simplistic growth analysis (on one or more planes on axial images) is able to properly capture aortic growth.

### 4.2. Limitations

Despite involving the largest uncomplicated type B AD population for CFD analysis to the authors’ knowledge, the sample size in this study is still unquestionably small, especially with such clear inter-patient variability and heterogeneity with respect to the time frame. Increasing the population size represents a challenge due to the relatively low incidence of the pathology under examination and the difficult segmentation process from poorly resolved images that often dictated the exclusion of a subject from the analysis. Non-gated CT scans provide a representation of the aortic anatomy as a temporal-weighted average over the cardiac cycle, therefore simultaneously capturing the position of the intimal flap at the diastolic and systolic configuration. The intimal flap motion, mostly evident in the acute phase of the disease, was detrimental to the image quality and the general segmentation in a few cases. Given the impact of segmentation on CFD errors and variability, the overall segmentation process represents a limitation especially when complex geometries and anatomical details are present.

Although most of the patients showed little to no flap motion, this characteristic points to the rigid wall assumption as another limitation of this study. The inclusion of a moving wall introduces a series of additional assumptions along with increased computational costs but could have important effects on the hemodynamic parameters of interest, mainly for those patients in an acute phase presenting a compliant flap.

The imaging aspect also represents a limitation in terms of applied boundary conditions. Despite the potential of phase-contrast MRI to provide patient-specific information at the boundaries [14,38], CT is currently the most used modality for the assessment of ADs in clinical settings.

## 5. Conclusions

The present study reports findings from CFD analysis on a population of residual type B aortic dissection patients with known outcomes. The results highlight the importance of considering geometric and hemodynamic parameters, as well as their interplay, when assessing individual aortas for prognostic purposes. In doing so, the CFD approach provides a reliable tool to estimate highly resolved patient-specific fluid dynamic parameters.

Rapid aortic expansion was found to be associated with larger FL. A large FL size, along with a large entry tear, is likely to promote FL patency, highlighting the strong dependence and interplay between geometry and hemodynamics. High FL flow rate and tortuosity were associated with adverse outcome (reported death during follow-up) and emerged as indicators of risk.

An aortic dissection induces complex changes in the vessel geometry and hemodynamics that result in extreme variability among patients, often increased by the presence of comorbidities or previous aortic surgeries. If uncomplicated type B AD constitutes a subgroup among type B AD patients, residual type B dissections are likely to define a subcategory in the uncomplicated group. Although more investigations are required, the reported findings emphasize the need for a patient-tailored approach when evaluating uncomplicated type B AD patients and show the potential of CFD-derived hemodynamics to complement anatomical assessment and help patient management.

## Figures and Tables

**Figure 1 bioengineering-11-00690-f001:**
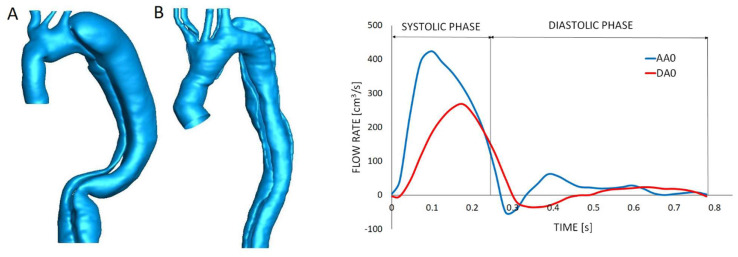
(**Left**): three-dimensional reconstruction of the aortic lumen for two example geometries; patient A with rapidly growing aorta and patient B with stable aortic size. (**Right**): flow rate extracted from PC-MRI images on a plane including the ascending and descending aorta.

**Figure 2 bioengineering-11-00690-f002:**
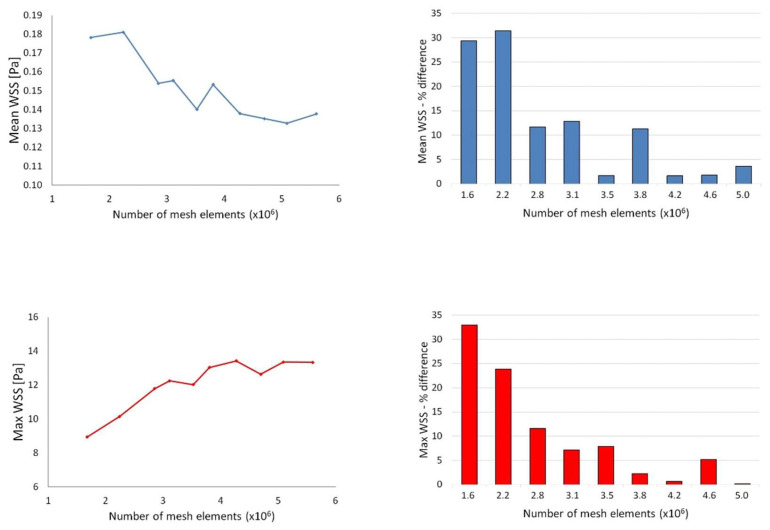
Sensitivity analysis for a case example. (**Top**): mean WSS as a function of the number of elements in the mesh (**Left**) and mean WSS relative percentage difference for each mesh with respect to the finer mesh (**Right**). (**Bottom**): maximum WSS as a function of the number of elements in the mesh (**Left**) and the maximum WSS relative percentage difference for each mesh with respect to the finer mesh (**Right**).

**Figure 3 bioengineering-11-00690-f003:**
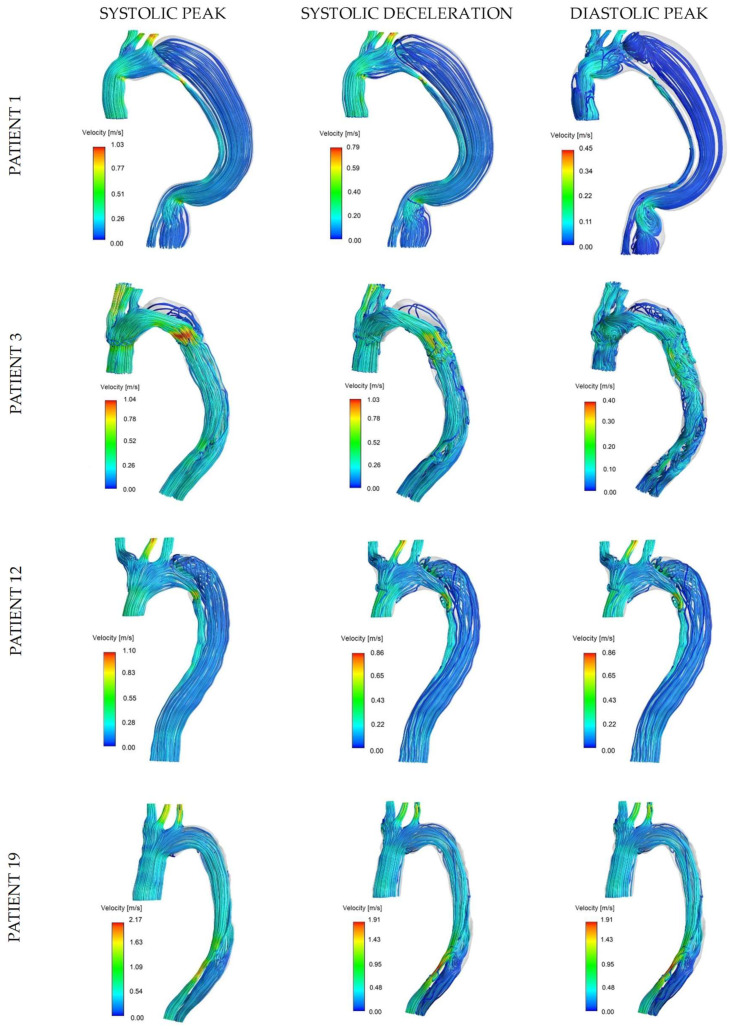
Flow features in the dissected aorta. Streamlines for the example cases at three instances of the cardiac cycle. Patients 1 and 3 belong to the rapid-growth group, patient 12 to the stable-size group/favorable outcome subgroup, and patient 19 to the stable-size group/adverse-outcome subgroup.

**Figure 4 bioengineering-11-00690-f004:**
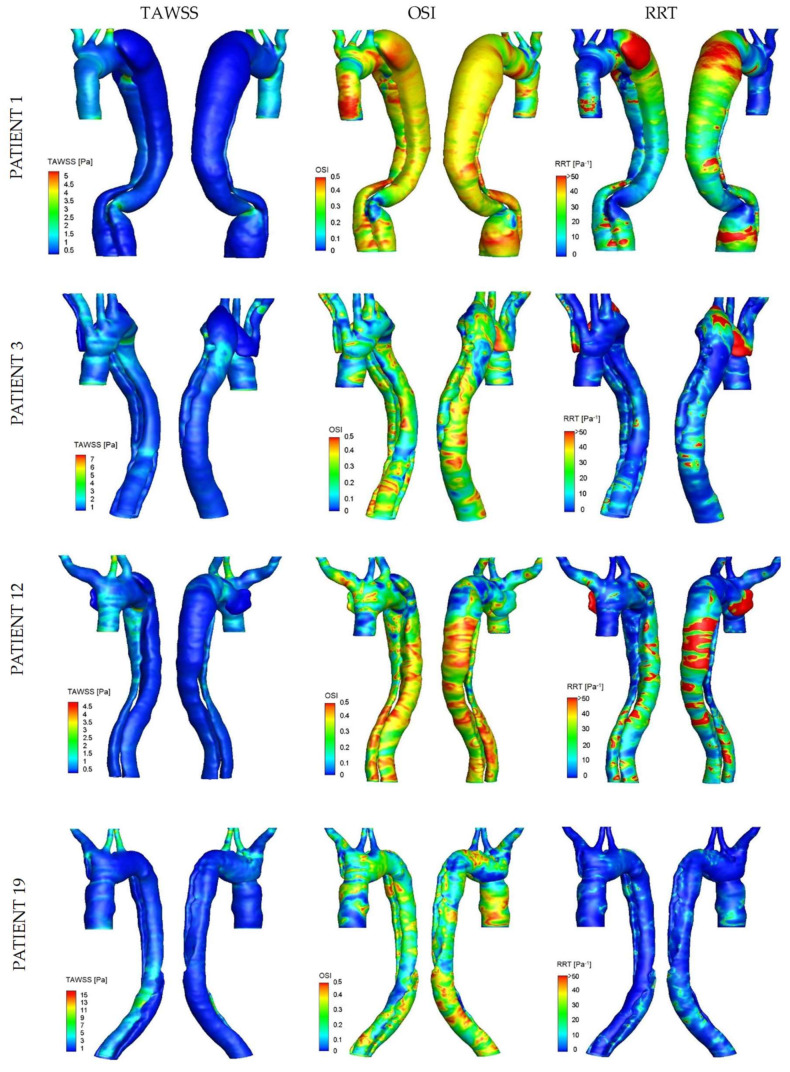
Luminal distribution of TAWSS, OSI, and RRT: example cases. Patients 1 and 3 belong to the rapid-growth group, patient 12 to the stable-size group/favorable outcome subgroup, and patient 19 to the stable-size group/adverse-outcome subgroup.

**Figure 5 bioengineering-11-00690-f005:**
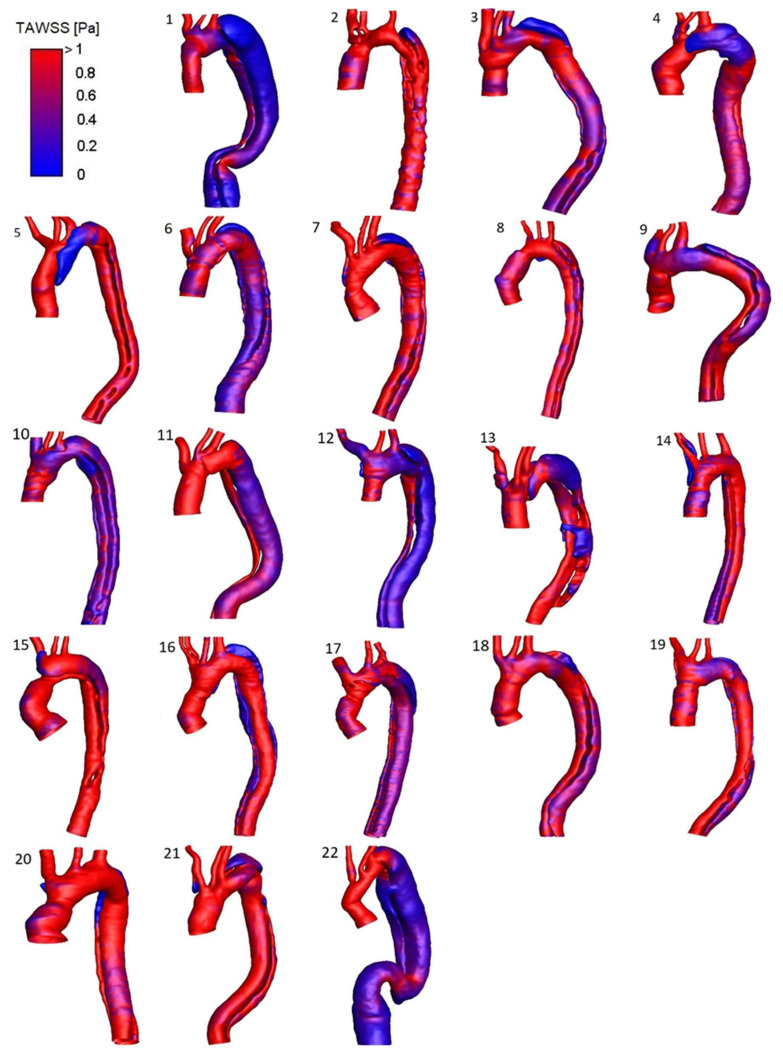
Luminal distribution of TAWSS for all the AD anatomies in the study population: patients 1–11 belong to the rapid-growth group, patients 12–22 to the stable-size group with patients 12–18 belonging to the favorable outcome subgroup, and patients 19–22 to the adverse-outcome subgroup.

**Figure 6 bioengineering-11-00690-f006:**
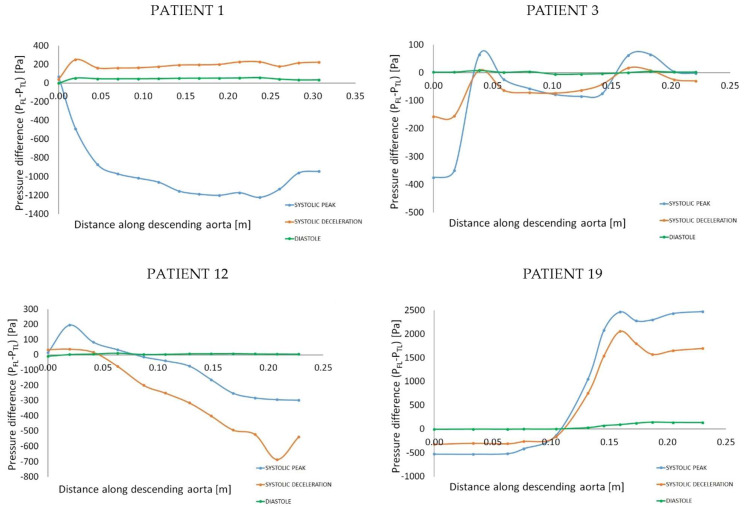
Pressure difference between the false and true lumens: differences were obtained as average FL pressure minus average TL pressure on cross-sectional planes perpendicular to the vessel’s centerline along the length of the descending aorta. The difference is reported at systolic peak, systolic deceleration, and diastole for the four representative examples. TOP row from the left: patients 1 and 3 belong to the rapid-growth group; BOTTOM row from the left: patient 12 belongs to the stable-size group/favorable outcome subgroup and patient 19 to the stable-size group/adverse-outcome subgroup.

**Table 1 bioengineering-11-00690-t001:** Summary of the geometric and hemodynamic variables used for comparison between the stable-size group and the rapid-growth group of residual type B aortic dissections. Mean values with standard deviation are reported along with two-tailed *p*-values. Statistically significant *p*-values are bold.

	Stable Size (n = 11)	Rapid Growth (n = 11)	*p*-Value
Age	60.6 ± 14.6	53.0 ± 10.3	0.192
Male sex	6 (55%)	9 (82%)	0.400
Initial aortic diameter [mm]	35.2 ± 11.1	35.5 ± 7.5	0.237
TL/DA area ratio	0.4 ± 0.1	0.3 ± 0.1	**0.009**
Entry tear size [mm^2^]	144.8 ± 173.7	210.0 ± 182.6	0.301
Number of re-entry tears	2.1 ± 1.6	1.5 ± 1.4	0.415
Tortuosity	1.4 ± 0.3	1.4 ± 0.3	0.995
FL flow [%]	55 ± 20	63 ± 15	0.331
Peak TAWSS (entry tear) [Pa]	7.9 ± 4.5	10.9 ± 9.4	0.308
TAWSS × margins (entry tear) [Pa∙mm]	255.6 ± 220.5	341.8 ± 189.1	0.134

**Table 2 bioengineering-11-00690-t002:** Summary of the geometric and hemodynamic variables used for comparison between the two subgroups of the stable-size residual type B aortic dissections. The subgroups were defined according to favorable or adverse outcome (death). Mean values with standard deviation are reported along with two-tailed *p*-values. Statistically significant *p*-values are bold.

	Stable Size— Favorable Outcome (n = 7)	Stable Size— Adverse Outcome (n = 4)	*p*-Value
Age	59.0 ± 10.1	62.8 ± 13.6	0.736
Male sex	3 (43%)	3 (75%)	0.545
Initial aortic diameter [mm]	32.5 ± 4.6	39.9 ± 17.9	0.527
TL/DA area ratio	0.4 ± 0.1	0.5 ± 0.1	0.472
Entry tear size [mm^2^]	67.1 ± 49.3	280.7 ± 238.7	**0.015**
Number of re-entry tears	2.4 ± 1.5	1.5 ± 1.5	0.461
Tortuosity	1.3 ± 0.2	1.5 ± 0.4	0.274
FL flow [%]	50 ± 16	66 ± 24	0.265
Peak TAWSS (entry tear) [Pa]	6.7 ± 4.5	10.1 ± 4.1	0.251
TAWSS × margins (entry tear) [Pa∙mm]	143.4 ± 91.3	452.0 ± 254.3	**0.022**

## Data Availability

Data are contained within the article.

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
