# Peer review of "Predicting Aneurysmal Degeneration in Uncomplicated Residual Type B Aortic Dissection"

_bioengineering, 2024, doi:10.3390/bioengineering11070690_

Round 1
Reviewer 1 Report
Comments and Suggestions for Authors
This artice tried to assess individual aortas for a population of type B ADs, and analyze potential correlations with known outcomes, such as rapid aortic growth indicating aneurysmal degeneration in terms of hemodynamcis and geometric features. The topic is quite interesting but I have the following major concerns:
1. The hemodynamics in FL is highly depend on the outlet boundary conditions, in this study, all the FLs are truncated which obviously is not true. Most FLs are closed lumen with flow entering and leaving through the tears, that’s why not only the proximol tears are important but distal tears also need to be accounted for . and the inapporpriate simuilation may lead to complete wrone results. For example, in this study all the distal FLs remained to be patent, in reality some FLs should become thrombosed.
2. It will be good to include more information about the distal tears, including tear amounts, the distance between tears etc.
Author Response
RESPONSE TO REVIEWER 1
Thank you for taking the time to review our manuscript and provide comments and feedback. Please find below the response to your review and the revised version of the manuscript.
- The hemodynamics in FL is highly depend on the outlet boundary conditions, in this study, all the FLs are truncated which obviously is not true. Most FLs are closed lumen with flow entering and leaving through the tears, that’s why not only the proximol tears are important but distal tears also need to be accounted for . and the inapporpriate simuilation may lead to complete wrone results. For example, in this study all the distal FLs remained to be patent, in reality some FLs should become thrombosed.
The authors appreciate this comment and agree on the importance of appropriate geometric reconstruction to obtain the appropriate computational domain. All the aortas in the study were carefully reconstructed from CT images and all the visible entry and exit tears were included in the segmentation and geometry reconstruction. Most of the geometry presented indeed a patent FL that was truncated just above the abdominal aorta in order to consistently use the thoracic aorta as computational domain. The visualization of the complete geometry is limited in the figures included in the manuscript so not all the images allow the visualization of the different tears, but on a number of patients we can appreciate the presence of re-entry tears (for example figure 2 and 4 patient 2, 3, 5, 10, 13, 15, 19). Similarly, a few patients present a closed FL (for example figure 4 patient 2, 13, 15, 22). As mentioned by the reviewer, the FL will become thrombosed in a number of cases, however the cases included in the study, and specifically the scan and time point used for the simulations did not show a thrombosed FL. The presence of thrombus in the FL at the baseline scan was used as an exclusion criterion for the study (please refer to the revised manuscript page 3, lines 119-128, reported below).
Given the retrospective nature of this study, only patients with at least one-year follow-up, or urgent reintervention within a year, were selected, for a total of forty-one subjects. The following exclusion criteria were then applied: CT scans with poor contrast/resolution, ex-tensive presence of thrombus in the descending aorta, marked intimal flap motion when detrimental to image quality and identification of anatomical features, presence of intra-mural hematoma or penetrating aortic ulcer, connective tissue disorders (i.e. Marfan syn-drome and Loeys-Dietz syndrome), other underlying conditions (i.e. giant cell aortitis), lack of immediate scan after surgery for type A AD, and previous intervention in the descending aorta.
- It will be good to include more information about the distal tears, including tear amounts, the distance between tears etc.
A sentence was added to the revised version of the manuscript (page 7, lines 260-262 and 268-272) to provide more information on distal tears for the aortic geometries in the study population.
The stable size group presented more re-entry tears on average but did not show a significant difference when compared to the rapid growth group (stable size 2.1±1.6 - rapid growth 1.5±1.4; p = .415).
The two subgroups defined according to long-term outcome of subjects in the stable size group did not differ significantly in initial maximum diameter (favorable outcome 32.49 ± 4.63 mm - adverse outcome 39.92 ± 17.92 mm; p = .527) or , true lumen size as ratio of total aortic size (favorable outcome 0.40 ± 0.11 - adverse outcome 0.46 ± 0.13; p = .472), or number of re-entry tears (favorable outcome 2.4±1.5 - adverse outcome 1.5±1.5; p = .461).
Reviewer 2 Report
Comments and Suggestions for Authors
There is a considerable proportion of patients with type B aortic dissection (AD) experience long-term complications, including the creation of an aneurysm in the false lumen (FL). The key to defending the hazards of interventional therapy is the capacity to identify patients who are more likely to develop aneurysms. By directing type B dissection illness management, the study of patient-specific hemodynamics may make it possible to implement a patient-tailored approach that improves prognosis.
This study analyzed group of patients with residual type B AD, individual aortas were evaluated retrospectively, and relationships with known outcomes (such as mortality or rapid aortic expansion) using CFD-derived hemodynamic descriptors and geometric features.
TAWSS at the tear site has been proposed as a potential measure of lumen-to-lumen dynamics and its impact on the development of individual aortas. Elevated FL flow rate and tortuosity were linked to unfavorable results, implying their potential as risk markers.
The paper is nice written and well arranged.
1. Mesh analisys: please provide mesh convergence plot and mesh statistics (number of nodes and elements)
2. Not much cases considered. Please insert a section in Discussion and make some plots showing how your results are correspond to reality within such small statistics (only 4 unfavourable cases were discussed). I suggest to include more cases. You can use literature data if you do not have enough clinical data.
3. Please divide 2.3. Computational Simulations into sub-paragraphs with title sections
Author Response
RESPONSE TO REVIEWER 2
Thank you for taking the time to review our manuscript and provide comments and feedback. Please find below the response to your review and the revised version of the manuscript.
- Mesh analisys: please provide mesh convergence plot and mesh statistics (number of nodes and elements)
The plots for a representative sensitivity analysis for a case in the study population was added to the revised manuscript (figure 2, page3-4, lines 148-158, reported below).
Consecutive grid refinements were used to obtain a range of mesh sizes from coarse (approximately 1 million elements) to fine (approximately 5 million elements), for a total of ten different meshes for each case. Steady-state simulations were then run at the cycle-averaged flow rate to obtain mean and maximum WSS values for each mesh. The effect of mesh density on the variables of interest was assessed by means of relative percentage difference with respect to the finer (more accurate) mesh. The results from a representative sensitivity analysis are reported in figure 2: the effect of mesh refinement on the mean and maximum WSS were negligible for a mesh density of about 3 million tetrahedral elements for this case. The final mesh density for the population ranged between 3 and 5 million elements de-pending on individual aortic size and morphology.
- Not much cases considered. Please insert a section in Discussion and make some plots showing how your results are correspond to reality within such small statistics (only 4 unfavourable cases were discussed). I suggest to include more cases. You can use literature data if you do not have enough clinical data.
The authors agree that one limitation of the research paper is the small sample size (please refer to the revised manuscript page 18, lines 563-568, reported below).
Despite involving the largest uncomplicated type B AD population for CFD analysis to the authors’ knowledge, the sample size in this study is still unquestionably small, especially with such clear inter-patient variability and heterogeneity with respect to time frame. Increasing the population size represents a challenge due to the relatively low incidence of the pathology under examination and the difficult segmentation process from poorly resolved images that often dictated the exclusion of a subject from the analysis.
The goal of the study was to analyze potential correlations between the hemodynamics of individual aortas and outcome, specifically aortic growth indicating aneurysmal degeneration. Eleven patients (50%) were found to have unfavorable clinical outcome (i.e. rapid aortic growth). Within the patients with stable aortic size, 4 patients were found to have adverse long-term outcome such as reintervention and death despite having a stable aortic size. For this reason, the authors decided to report and comment on this smaller subgroup that appears to have clinical relevance. It should be noted that the main clinical outcome considered in the study was aortic growth, with a total of 11 unfavorable cases discussed.
The authors agree that it would be interesting to include more cases; however, these were the patients that could be analyzed based on inclusion criteria and ethic protocol. The study focused on patients presenting residual type B aortic dissection after receiving hemi-arch repair for type A aortic dissection. We thoroughly analyzed the retrospective cohort at our disposal and no additional patients with baseline and follow-up CT scans for residual type B aortic dissection was available to run CFD simulations and assess growth consistently with our research protocol. Using literature data would be very risky and misleading for a study involving computational simulations. A different CFD study for aortic dissections could present different boundary conditions, simulations setup, etc., making comparisons difficult and scientifically poor. Additionally, the assessment and definition of rapid aortic growth could potentially also differ in cases obtained from literature. To the best of the authors’ knowledge this is the largest uncomplicated type B AD population for a CFD study.
- Please divide 2.3. Computational Simulations into sub-paragraphs with title sections
Subsection 2.3. was divided into 3 subsubsections, namely 2.3.1. Boundary Conditions, 2.3.2. Blood Rheology and Fluid Flow Regime, 2.3.3. Numerical Implementation.
Round 2
Reviewer 2 Report
Comments and Suggestions for Authors
There is a considerable proportion of patients with type B aortic dissection (AD) experience long-term complications, including the creation of an aneurysm in the false lumen (FL). The key to defending the hazards of interventional therapy is the capacity to identify patients who are more likely to develop aneurysms. By directing type B dissection illness management, the study of patient-specific hemodynamics may make it possible to implement a patient-tailored approach that improves prognosis.
This study analyzed group of patients with residual type B AD, individual aortas were evaluated retrospectively, and relationships with known outcomes (such as mortality or rapid aortic expansion) using CFD-derived hemodynamic descriptors and geometric features.
TAWSS at the tear site has been proposed as a potential measure of lumen-to-lumen dynamics and its impact on the development of individual aortas. Elevated FL flow rate and tortuosity were linked to unfavorable results, implying their potential as risk markers.
Paper has become better.
One small comment:
1) Please provide along Fig. 7, the numerical results suggesting these clinical data. It would be nice. Fig. 7 alone is not well looking here.
Author Response
Thank you for taking the time to review our manuscript and provide comments and feedback. Please find below the response to your review and the revised version of the manuscript.
1) Please provide along Fig. 7, the numerical results suggesting these clinical data. It would be nice. Fig. 7 alone is not well looking here.
Please note that figure 7 is used in the context of the manuscript discussion in order to support an hypothesis on the results for the small adverse outcome subgroup (n= 4 patients), i.e. the patients the presented adverse clinical outcome despite showing a stable aortic size. In this paragraph each patient in the subgroup is discussed separately to try and hypothesize the reason for the adverse clinical outcome. Specifically we think that patient 22 could represent a case of a dissection at a more advanced stage compared to the rest of the population, also given the very short surveillance time between baseline scan and follow up scan (5 months) before re-intervention and death. For this reason this patient may not have shown growth, given the very short surveillance time, but presented adverse outcome. The purpose of figure 7 is to support this idea by showing a very defined aortic geometry with dissection, with clear thrombus and no intimal flap motion, usually sings of a chronic, i.e. more advanced stage of the aortic dissection. The numerical results for this case are reported in the results section along with the rest of the population under investigation. There are no specific numerical data directly related to figure 7, this image of a patient's CT scan is used to support an hypothesis for this patient in the context of the subgroup with stable size and adverse clinical outcome.